

# Θ-SEIHRD mathematical model of Covid19-stability analysis using fast-slow decomposition

OPhir Nave[1], Israel Hartuv[2] and Uziel Shemesh[2]

[1] Department of Mathematics, Jerusalem College of Technology, Jerusalem, Israel
[2] Department of Computer Science, Jerusalem College of Technology, Jerusalem, Israel

## ABSTRACT

In general, a mathematical model that contains many linear/nonlinear differential equations, describing a phenomenon, does not have an explicit hierarchy of system variables. That is, the identification of the fast variables and the slow variables of the system is not explicitly clear. The decomposition of a system into fast and slow subsystems is usually based on intuitive ideas and knowledge of the mathematical model being investigated. In this study, we apply the singular perturbed vector field (SPVF) method to the COVID-19 mathematical model of to expose the hierarchy of the model. This decomposition enables us to rewrite the model in new coordinates in the form of fast and slow subsystems and, hence, to investigate only the fast subsystem with different asymptotic methods. In addition, this decomposition enables us to investigate the stability analysis of the model, which is important in case of COVID-19. We found the stable equilibrium points of the mathematical model and compared the results of the model with those reported by the Chinese authorities and found a fit of approximately 96 percent.

## INTRODUCTION

The coronavirus belongs to the severe acute respiratory syndrome (SARS) family. It usually does not cause disease in humans, but infects animals—mammals and poultry. If humans are infected with the virus, the disease usually causes mild cooling, and the infection passes without any treatment. Conversely, if the infected person has a weak immune system for some reason, the coronavirus can be fatal. It is a common virus that attacks almost every person at least once in his/her lifetime—especially during childhood — and, as is usually the case, it is not dangerous. However, the current epidemic is a recently mutated virus that has become fatal (which is why the World Health Organization initially called it "the new coronavirus" 2019). It should be noted that, in 2002–2003, there was an outbreak of a similar virus from the SARS family, causing the death of more than 800 people (General Health Fund in Israel, https://www.clalit.co.il/he/yourhealth/family/

Corresponding author
OPhir Nave, naveof@gmail.com

Pages/coronavirus.aspx). The virus passes from person to person with a drop infection—as in the case of influenza: droplets from an infected person carry the virus into the air and penetrate the respiratory system of uninfected humans. The virus can also be passed through contact with an infected person. Hence, the importance of frequent hand washing with water and soap or alcohol is propagated. This is also why shaking hands and touching the face are to be avoided. It should be noted that the infected person can be either symptomatic (has the symptoms of the disease), pre-symptomatic (infected but has yet to develop symptoms), or asymptomatic (infected with the virus, but has no symptoms and will not develop the disease). By now, we already know that a significant proportion of those infected are asymptomatic and, therefore, without being aware of it, can endanger those who are not infected. Hence, the directive that everyone, without exception, should adhere to the guidelines on maintaining social distancing. In light of the current situation, and in the absence of a scientifically proven drug or vaccine, scientists from all over the world have been trying to find an effective vaccine for this virus. They include researchers from various fields of science such as biology, chemistry, physics and even mathematics. Researchers have developed various mathematical models that describe the evolution of the virus in the population (*Liu et al., 2020*; *Ivorra & Ramos, 2020a*, *2020b*; *Yan & Cao, 2019*; *Kucharski et al., 2020*; *Li et al., 2020*; *Ivorra et al., 2014*; *Johns Hopkins University (JHU), 2020*). All countries whose population was infected with the virus have tried to prevent its spread by imposing a complete lockdown of the population. This effort is an attempt to contain the occurrence of new infections. The mathematical description for this situation in which there are no new infections, which means that the spread has stopped, is called equilibrium. Therefore, the optimal state is equilibrium stability. In this paper, we investigate the stability of the $\theta$-SEIHRD model, a mathematical model proposed by *Ivorra et al. (2020)*.

In general, given a large mathematical model that contains a system of linear or nonlinear differential equations, it is difficult, and sometimes impossible, to analyze the model analytically. Even numerical analysis is sometimes complicated and requires a very expensive computer running time. Therefore, the need to reduce the number of equations (i.e., reduced model) while maintaining the physical/biological dynamics of the system is required. In this paper, we introduce and apply a method called singular perturbed vector field (*SPVF*) (*Bykov, Goldfarb & Gol'dshtein, 2006*; *Nave, 2019*), which is a new version of the *ILDM* method. Because a large number of differential equations (a mathematical model) are presented in a hidden hierarchy form, there is no explicit time scale of the system. Hence, our aim is to first expose the time scale of a given system and then apply a reduction method. This is the main result of the *SPVF* method. The *SPVF* method transfers the original system to the form of the singularly perturbed system (*SPS*), that is, a system with an explicit hierarchy of the dynamic variables of the model. Once we transfer the system to an *SPS* one, the new system can be treated by the highly effective standard *SPS* theory for model reduction and decomposition, as we mentioned above, without losing the essential dynamics of the original system.

The rest of the paper is organized as follows. In "The θ-SEIHRD Model of COVID-19", we describe the mathematical model of the coronavirus called θ-SEIHRD model. In "Results and Analysis", we present the results of our comparative analysis of the cases reported by the Chinese authorities. In addition, we apply the SPVF model to the θ-SEIHRD model and find the stability of the equilibrium points. "Discussion" presents our discussion. Finally, "Conclusions" presents the conclusion of this paper.

## THE Θ-SEIHRD MODEL OF COVID-19

In this section, we introduce the mathematical model of the θ-SEIHRD model as presented in (*Ivorra et al., 2020*). This model is based on the *Be−CoDiS* model presented in (*Ivorra, Ramos & Ngom, 2015*). It is important at this point to note that this model is not the SIR-SEIR standard model. It considers the known special characteristics of the considered disease, such as the existence of infectious undetected cases and the sanitary conditions and infectiousness of hospitalized patients. The assumptions of the model can be found in (*Ivorra et al., 2020*). Here, we present the mathematical formulation of the COVID-19 spread. The system of equations includes nine nonlinear ordinary differential equations and has the following forms:

$$\frac{dS}{dt} = -\frac{S}{N}\left(m_{\varepsilon}\beta_E E + m_I \beta_I I + m_{I_u}\beta_{I_u} I_u\right) - \frac{S}{N}\left(m_{H_R}\beta_{H_R} H_R + m_{H_D}\beta_{H_D} H_D\right)$$
$$- \mu_m S + \mu_n(S + E + I + I_u + R_d + R_u) \equiv F_S(\vec{V}) \tag{1}$$

$$\frac{dE}{dt} = \frac{S}{N}\left(m_{\varepsilon}\beta_E E + m_I \beta_I I + m_{I_u}\beta_{I_u} I_u\right) + \frac{S}{N}\left(m_{H_R}\beta_{H_R} H_R + m_{H_D}\beta_{H_D} H_D\right)$$
$$- \mu_m E - \gamma_{\varepsilon} E + \tau_1 - \tau_2 \equiv F_E(\vec{V}) \tag{2}$$

$$\frac{dI}{dt} = \gamma_{\varepsilon} E - \left(\mu_m - \gamma_I\right) I \equiv F_I(\vec{V}) \tag{3}$$

$$\frac{dI_u}{dt} = (1 - \theta)\gamma_I I - \left(\mu_m + \gamma_{I_u}\right) I_u \equiv F_{I_u}(\vec{V}) \tag{4}$$

$$\frac{dH_R}{dt} = \theta\left(1 - \frac{\omega(t)}{\theta}\right)\gamma_I I - \gamma_{H_R} H_R \equiv F_{H_R}(\vec{V}) \tag{5}$$

$$\frac{dH_D}{dt} = \omega(t)\gamma_I I - \gamma_{H_D} H_D \equiv F_{H_D}(\vec{V}) \tag{6}$$

$$\frac{dR_d}{dt} = \gamma_{H_R} H_R - \mu_m R_d \equiv F_{R_d}(\vec{V}) \tag{7}$$

$$\frac{dR_u}{dt} = \gamma_{I_u} I_u - \mu_m R_u \equiv F_{R_u}(\vec{V}) \tag{8}$$

$$\frac{dD}{dt} = \gamma_{H_D} H_d \equiv F_D(\vec{V}) \tag{9}$$

where $\vec{V} = (S, E, I, I_u, H_R, H_D, R_d, R_u, D)$; $\gamma_E$, $\gamma_I$, $\gamma_{H_R}$ and $\gamma_{I_u}$ are the transition rates of $E$, $I$, $H_R$, and $I_u$, respectively $\left[\frac{1}{day}\right]$; $\beta(\cdot)$ is the disease contact rate $\left[\frac{1}{day}\right]$; $\mu$ is the fatality rate $\left[\frac{1}{day}\right]$; and $\tau$ indicates the infected people, who move from one territory to another per day. The initial conditions of the system at time $t_0$ (for each country, however, the initial time can be changed. For example, in Wuhan, China, $t_0 = 10$ *A. CET* (technical adjustment) because it was on 7 December 2019 when the first case of patients with symptoms was confirmed) are as follows:

$$S(t_0) = S_0, \quad E(t_0) = E_0, \quad I(t_0) = I_0, \quad I_u(t_0) = I_{u0}, \quad H_R(t_0) = H_{R0},$$
$$H_D(t_0) = H_{D0}, \quad R_d(t_0) = R_{d0}, \quad R_u(t_0) = R_{u0}, \quad D(t_0) = D_0 \tag{10}$$

As can be seen above, the presentation of the above mathematical model has no explicit hierarchy and, hence, it is impossible to apply different asymptotic methods. In the next section, we apply the SPVF methods to expose the hierarchy of the considered model.

## RESULTS AND ANALYSIS

In this section, we apply the SPVF method to the system of Eqs. (1)–(9), as presented in (*Nave, 2017*). While we apply the SPVF method, we expose the hierarchy of the model. This procedure enables us to rewrite the model under consideration in new coordinates and, hence, the "new" model can be decomposed into the so-called fast and slow subsystems.

The set of parameters and functions used in our calculations is as follows:

### Parameters:

$$\gamma_E = 0.1818, \quad \beta_E = 0.32, \quad \gamma_I \in [0.1493, 1.4286], \quad \gamma_{H_R} \in [0.0752, 0.1370],$$
$$\gamma_{I_u} \in [0.0752, 0.1370], \quad \beta_I = 0.2887, \quad \beta_{I_u} \in [0.1773, 0.2701],$$
$$\beta_{H_R} \in [0.0034, 0.0131], \quad \beta_{H_D} \in [0.0034, 0.0131], N = 1000, \quad \mu_m = \mu_n = 1,$$
$$\tau_1 = \tau_2 = 1, \quad d_{I_u} = 7.3, \delta_R = 18, \quad \theta = 0.14 \tag{11}$$

### Functions:

$$\beta_{I_u} = 0.375 + \frac{0.375}{(1 - \omega(t))}(1 - \theta), \omega(t) = 0.003 \cdot m_I + 0.997 \cdot (1 - m_I),$$
$$m_E = m_{H_R} = m_{H_D} = m_{I_u} = m_I = e^{-0.01(t-23)}, \quad \gamma_{H_D} = \frac{1}{d_{I_u} + 6(1 - m_I) + \delta_R} \tag{12}$$
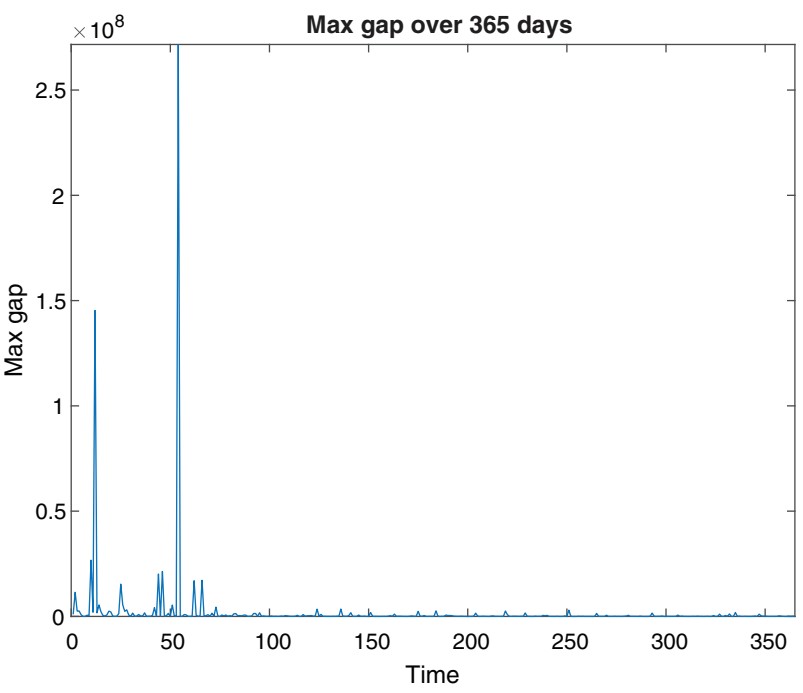

**Figure 1 Maximal Gap.** The MaxGap for every day during 365 days.

Here, we present only the absolute values of the eigenvalues of the SPVF method (and not the eigenvectors) to avoid overwhelming the readers with too much information:

$$\lambda_1 = 2.1356 \cdot 10^{25}$$
$$\lambda_2 = 7.8618 \cdot 10^{16}$$
$$\lambda_3 = 5.3590 \cdot 10^{16}$$
$$\lambda_4 = 3.4835 \cdot 10^{15}$$
$$\lambda_5 = 1.0870 \cdot 10^{15}$$
$$\lambda_6 = 1.0870 \cdot 10^{15}$$
$$\lambda_7 = 5.4587 \cdot 10^{14}$$
$$\lambda_8 = 5.4586 \cdot 10^{14}$$
$$\lambda_9 = 3.8120 \cdot 10^{13} \tag{13}$$

As we observed from the above results, and according to the SPVF method, the maximal gap of the eigenvalues is between $\lambda_1$ and $\lambda_2$. This gap implies that the first dynamic variable of the system, which is written in the new coordinates, is fast compared with the rest of the variables.

The results of the SPVF algorithm are presents in Figs. 1 and 2. We compute the value of $|\lambda_{i+1}|/|\lambda_i|$ ($i = 1,\dots,9$), for 365 days and plot only the maximum values of this quotient for every day. According to Fig. 1 we can see that for every day the gap indeed exists (for every day the graph is not zero). This means that the SPVF algorithm exposes the hierarchy of the system at the new coordinates. But it is not clear from these results what is the
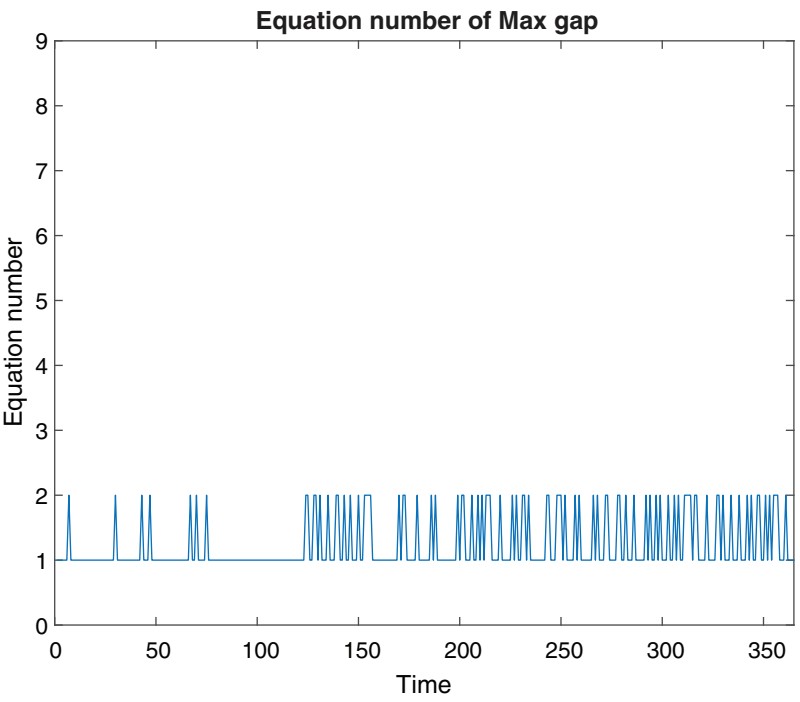

**Figure 2 Fast variables\equation.** The fast subsystem according to the decomposition of the SPVF method.

fast subsystem and what is the slow one. Hence, in order to know what is the fast subsystem and what is the slow one we match every eigenvalue to its corresponding equation at the model such that the eigenvalue $\lambda_i$ belongs to the $i$ equation at the new coordinates. Once we did this matching and with the results of the maximal gap (present at Fig. 1) we can know what is the fast subsystem and what is the slow one. For example, if the maximal gap is $|\lambda_6|/|\lambda_7|$ then the fast subsystem is Eqs. (1)–(6) and the slow subsystem are Eqs. (7)–(9). The results are present in Fig. 2. We can see that at some days the fast subsystem includes only the first equation and for some other days the fast subsystem includes the first two equations of the model in the new coordinates. One can analyze the results of the SPVF algorithm from the other direction. That is, if we look at Fig. 2 on the 100th day, for example, then we see that only the first equation is the fast subsystem. But if we look at the 250th day for example, then the first two equations are the fast subsystem.

The SPVF algorithm aims to find a frame of coordinates for which the system has an SPS form. Hence, we have a degree of freedom to choose any frame of coordinates for this purpose. At our analysis, we have chosen the frame of coordinates, that is, eigenvectors, that belong to the 49th day where on this day we received the maximal gap of all the gaps that we obtained from the SPVF algorithm. On this day the fast subsystem is the first equation of the model.

We denote the new variables of the model in the new coordinates as

$$\tilde{\vec{V}} = (\tilde{S}, \tilde{E}, \tilde{I}, \tilde{I}_u, \tilde{H}_R, \tilde{H}_D, \tilde{R}_d, \tilde{R}_u, \tilde{D}) \qquad (14)$$

The new model is rewritten in the new coordinates using the eigenvectors $\vec{u}_1, \vec{u}_2, \ldots, \vec{u}_9$, which correspond to the eigenvalues $\lambda_1, \lambda_2, \ldots, \lambda_9$. This means that the new dynamic variables of the model are linear combinations of its old variables when the coefficients of the linear combinations are taken from the eigenvectors, that is,

$$
\begin{pmatrix} \tilde{S} \\ \tilde{E} \\ \tilde{I} \\ \tilde{I}_u \\ \tilde{H}_R \\ \tilde{H}_D \\ \tilde{R}_d \\ \tilde{R}_u \\ \tilde{D} \end{pmatrix} = \begin{pmatrix} \vdots & \vdots & \vdots & \vdots & \vdots & \vdots & \vdots & \vdots & \vdots \\ \vec{u}_1^t & \vec{u}_2^t & \vec{u}_3^t & \vec{u}_4^t & \vec{u}_5^t & \vec{u}_6^t & \vec{u}_7^t & \vec{u}_8^t & \vec{u}_9^t \\ \vdots & \vdots & \vdots & \vdots & \vdots & \vdots & \vdots & \vdots & \vdots \end{pmatrix} \cdot \begin{pmatrix} S \\ E \\ I \\ I_u \\ H_R \\ H_D \\ R_d \\ R_u \\ D \end{pmatrix} \quad (15)
$$

where $t$ refers to the transpose operation. In matrix form, system (15) can be written as

$$
\tilde{\vec{V}} = \mathcal{B}\vec{V} \quad (16)
$$

Here, we denote the matrix $\mathcal{B}$ as a matrix whose columns are the eigenvectors of the SPVF method. The right-hand side (RHD) of this system is a function of the variables $\vec{V}$, whereas its left-hand side is a function of the new variables $\tilde{\vec{V}}$. To write the system of Eq. (16), with the same variables $\tilde{\vec{V}}$, we should differentiate this system with respect to time, that is,

$$
\dot{\tilde{\vec{V}}} = \mathcal{B}\dot{\vec{V}} = \mathcal{B}\vec{F}_{\vec{V}}(\vec{V}) \quad (17)
$$

where

$$
\vec{F}_{\vec{V}}(\vec{V}) = (F_S, F_E, F_I, F_{I_u}, F_{H_R}, F_{H_D}, F_{R_d}, F_{R_u}, F_D) \quad (18)
$$

is the vector field of systems (1)–(9). To rewrite both sides of the model using the new variables, we should express the old variables as a function of the new variables using the inverse matrix of the eigenvectors, that is,

$$
\mathcal{B}^{-1}\tilde{\vec{V}} = \vec{V} \quad (19)
$$

Substituting Eqs. (19) into (17), we obtain the mathematical model in the new coordinates with the initial conditions as

$$
\dot{\tilde{\vec{V}}} = \mathcal{B}\vec{F}_{\vec{V}}(\mathcal{B}^{-1}\tilde{\vec{V}}) \equiv H_{\tilde{\vec{V}}}(\tilde{\vec{V}})
$$

$$
\tilde{\vec{V}}(0) = \mathcal{B}\vec{V}(0) \quad (20)
$$

where

$$
H_{\tilde{\vec{V}}}(\tilde{\vec{V}}) = (H_{\tilde{S}}, H_{\tilde{E}}, H_{\tilde{I}}, H_{\tilde{I}_u}, H_{\tilde{H}_R}, H_{\tilde{H}_D}, H_{\tilde{R}_d}, H_{\tilde{R}_u}, H_{\tilde{D}}) \quad (21)
$$

is the vector field of the new model (the RHD of the model is written in the new coordinates).

## Stability analysis

After we transformed and presented the model in the new coordinates using the eigenvectors of the SPVF method, the model can be decomposed into the fast and slow subsystems based on the gap of the eigenvalues. The first eigenvalue, $\lambda_1$, indicate that the first variable of the system, say $\tilde{S}$, is the fast variable of the system and that $\tilde{E}, \tilde{I}, \tilde{I}_u, \tilde{H}_R, \tilde{H}_D,$ $\tilde{R}_d, \tilde{R}_u$ and $\tilde{D}$ are the slow ones. Hence, the model presented in the system of Eq. (20) can be decomposed as follows:

$$\text{fast subsystem} \begin{cases} \dfrac{d\tilde{S}}{dt} = H_{\tilde{S}}(\vec{V}) \end{cases}$$

$$\text{slow subsystem} \begin{cases} \dfrac{d\tilde{E}}{dt} = H_{\tilde{E}}\left(\vec{\tilde{V}}\right) \\ \dfrac{d\tilde{I}}{dt} = H_{\tilde{I}}\left(\vec{\tilde{V}}\right) \\ \dfrac{d\tilde{I}_u}{dt} = H_{\tilde{I}_u}\left(\vec{\tilde{V}}\right) \\ \dfrac{d\tilde{H}_R}{dt} = H_{\tilde{H}_R}\left(\vec{\tilde{V}}\right) \\ \dfrac{\tilde{H}_D}{dt} = H_{\tilde{H}_D}\left(\vec{\tilde{V}}\right) \\ \dfrac{\tilde{R}_d}{dt} = H_{\tilde{R}_d}\left(\vec{\tilde{V}}\right) \\ \dfrac{d\tilde{R}_u}{dt} = H_{\tilde{R}_u}\left(\vec{\tilde{V}}\right) \\ \dfrac{d\tilde{D}}{dt} = H_{\tilde{D}}\left(\vec{\tilde{V}}\right) \end{cases}$$

According to the SPVF method, the stability analysis can be investigated on the fast subsystem. To find the equilibrium points of the model, we should solve the following linear system:

$$H_{\tilde{S}}\left(\tilde{S}^*, \tilde{E}_0, \tilde{I}_0, \tilde{I}_{u0}, \tilde{H}_{R0}, \tilde{H}_{D0}, \tilde{R}_{d0}, \tilde{R}_{u0}\tilde{D}_0\right) = 0 \qquad (22)$$

This implies that we look for the equilibrium points of the fast variable (denoted by an asterisk), that is, $\tilde{S}^*$, while the rest of the variables are frozen, and we take their values as the initial conditions of the system. System (22) is a system of one equation with one unknown (fast) variables: $\tilde{S}^*$. We solve this system and substitute the values $\tilde{S}^*$ in the slow subsystem and solve the system for the slow variables, that is, $\tilde{E}^*, \tilde{I}^*, \tilde{I}_u^*, \tilde{H}_R^*, \tilde{H}_D^*, \tilde{R}_d^*, \tilde{R}_u^*$ and $\tilde{D}^*$. Now, we have eight equations with eight unknown (slow) variables.

To check the stability of these equilibrium points, we compute the Jacobian matrix calculated at the equilibrium point. It is important at this point to note that the equilibrium points in the new coordinates have no biological meaning. Moreover, to provide a biological interpretation to the equilibrium points that we received, we have to transfer
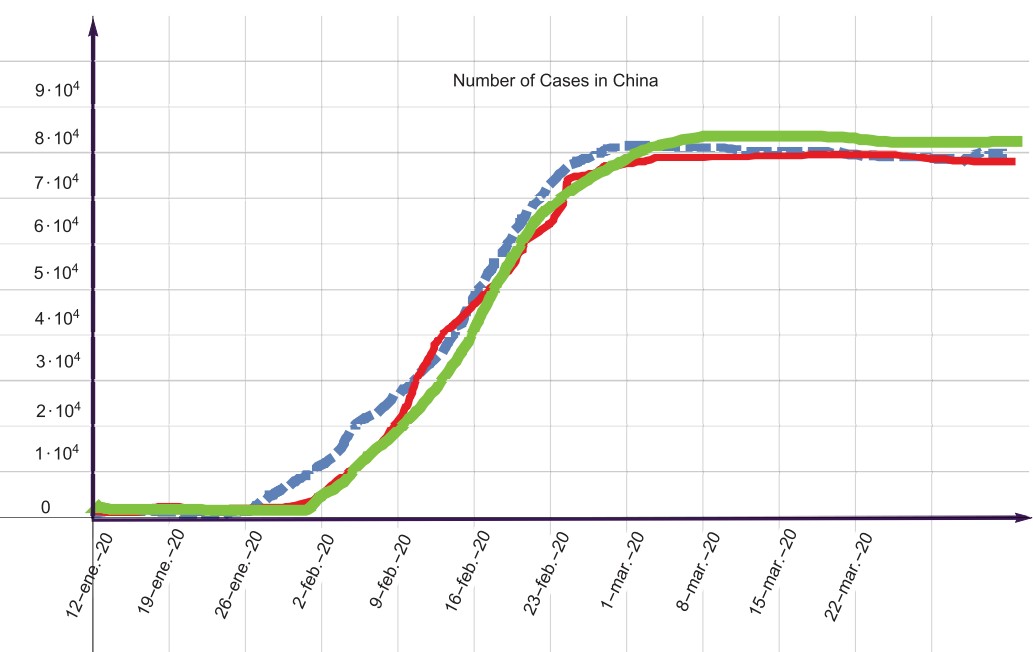

Figure 3 **Stability.** Solution profile of the number of cases of death in China.

Table 1 **Equilibrium point.** Equilibrium point of the model vs. Reporting from the authorities.

|  | Equilibrium point of the model | Reporting from the authorities |
| --- | --- | --- |
| Coronavirus Cases | 82,921 | 82,992 |
| Deaths | 4,597 | 4,634 |

them to the old coordinates for which we need to calculate the inverse matrix of the matrix with the eigenvectors, that is,

$$\vec{V}^*_{\text{stable}} = \mathcal{B}^{-1}\vec{\tilde{V}}^*_{\text{stable}} \tag{23}$$

Here, we inverse transform only the stable equilibrium points. We received only one stable point in the new coordinates, and, therefore, we should transform only one stable equilibrium point.

According to our numerical simulations, if the initial condition is on 12 January 2020, then the equilibrium point is reached after 49 days. According to official reports (from the Wuhan County authorities), the closure of Wuhan ended after 62 days. This means that the situation stabilized after 62 days. As we can see from the results of the mathematical model and what actually happened, there is a difference of 13 days. Medically, another 13 days of lockdown would not have affected the civilians. However, apparently, the economic considerations outweighed the medical considerations and the authorities decided to lift the lockdown earlier than expected. In Fig. 3, we present the evolution of the number of cases in China for different values of the parameters. As we can see from these results, the equilibrium point is attained at approximately 83$K$. The results are summarized in Table 1. In this table, we present only the important variables that are stable.

## DISCUSSION

As we have shown in the previous section, we obtain the stable equilibrium points of the mathematical model owing to the application of the SPVF method. The application of this method enables us to decompose the system into a fast and a slow subsystems. After the decomposition, we explored only the fast subsystem. In general, instead of decomposing a given system into fast and slow subsystems, one can solve the original system and determine the stable equilibrium points numerically. However, the big problem with the numerical method is that the equilibrium points are represented by their values and not as points, and they depend on the original system's parameters as can be obtained by the considered decomposition. That is, if we want to change the system parameters and determine new equilibrium points, we have to resolve the mathematical model numerically each time, which is time consuming. Moreover, the SPVF method allows us to first find all the equilibrium points of the original model analytically. In addition, the equilibrium points depend on the parameters of the original system. Therefore, if we want to change the model parameters, we do not have to resolve the mathematical model again but only need to change the parameters in the stable equilibrium points that have been determined. As can be seen from the results obtained from the mathematical model and from the results reported by the authorities, the relative error of the model can be calculated for which, we define the relative error of the equilibrium points for each dynamic variable of the system as

$$E_{(\cdot)} = \frac{|U_{\text{model}} - U_{\text{real}}|}{U_{\text{real}}} \cdot 100\% \tag{24}$$

where $U_{\text{model}}$ is the dynamic variable of the system, that is, the stable equilibrium point of the original model (after the inverse transform of the stable equilibrium points) and $U_{\text{real}}$ denotes the values reported by the authorities. The results are as follows:

$$E_S = 0.134\%, \quad E_E = 2.349\%, \quad E_I = 1.169\%, \quad E_{I_u} = 3.276\%, \quad E_{H_R} = 1.155\%,$$
$$E_{H_D} = 4.633\%, \quad E_{R_D} = 0.122\%, \quad E_{R_u} = 4.091\%, \quad E_D = 1.445\% \tag{25}$$

As can be seen from the results of the relative error, the reported cases are indeed small in number, as was expected from the decomposition method.

## CONCLUSIONS

In this paper, we investigated the stability of the $\theta$-SEIHRD mathematical model of COVID-19. The mathematical model of coronavirus is presented with a hidden hierarchy. This implies we cannot know which of the dynamic variables of the model is progressing fast and which one is progressing slowly. The hidden hierarchy of the model neither allows for an asymptotic analysis in general, nor an analytical investigation in particular, but only the running of numerical simulations. In particular, the equilibrium points of the system and their stability cannot be investigated and obtained from the present model. Therefore, in this study, we implemented the SPVF method, which explicitly exposes the system hierarchy. This method transforms the model into new coordinates using the

eigenvectors of the given vector field. In the new coordinates, the model is presented in the form of the SPS, that is, with an explicit hierarchy. The hierarchy of the new mathematical model enables us to decompose the model and divide it into subsystems. In our case, we only divided it into two subsystems: a fast and a slow subsystem. According to the SPVF theory, the fast subsystem can be investigated while the slow system is frozen. We found the equilibrium points the fast system in the new coordinates analytically. We analyzed the stability of the equilibrium points and obtained a single equilibrium point for the new model. To obtain a biological interpretation of the equilibrium point, we inverse transformed the stable equilibrium point into the old coordinates. According to our analytical results and the official reports of the Chinese authorities, we found that the stable equilibrium points obtained from the mathematical model are very close to those of the official reports.

## NOMENCLATURE

$S$      Susceptible: The person is not infected by the disease pathogen

$E$      Exposed: The person is in the incubation period after being infected by the disease pathogen and has no visible clinical signs. The individual could infect other people but with a lower probability than that by the people in the infectious compartments. After the incubation period, the person moves to the infectious compartment I

$I$      Infectious: After the incubation period, the first compartment of the infectious period in which nobody is expected to be detected yet. The person has completed the incubation period, may infect other people, and start developing clinical signs. After this period, people in this compartment can be either taken in charge by sanitary authorities (and we classify such people as hospitalized) or may not be detected by the said authorities and continue to be infectious (but in another compartment, $I_u$)

$I_u$      Infectious but undetected: After being in compartment I, the person can still infect other people and have clinical signs; however, they have not yet been detected and reported by the authorities. We assume that only people with low or medium symptoms can be included in this compartment, and not the people who die. After this period, people in this compartment move to the recovered compartment $R_u$.

$H_R$      Hospitalized: The person is in the hospital (or under quarantine at home) and can still infect other people. At the end of this stage, a person moves on to the Recovered compartment $R_d$

$H_D$      Hospitalized who will die: The person is hospitalized and can still infect other people. At the end of this state, the person is transferred to the Dead compartment

$R_d$      Recovered after being previously detected as infectious: The person was previously detected as infectious, survived the disease, is no longer infectious, and has developed a natural immunity to the virus. When a person enters this

compartment, he/she remains in the hospital for a convalescence period of $C_0$ days (average)

$R_u$        Recovered after being previously infectious but undetected: The person was not previously detected as infectious, survived the disease, is no longer infectious, and has developed a natural immunity to the virus

$D$        Dead by COVID-19: The person did not survive the disease

$t$        duration

### Funding
The authors received no funding for this work.

### Competing Interests
The authors declare that they have no competing interests.

### Author Contributions
- OPhir Nave conceived and designed the experiments, performed the experiments, analyzed the data, prepared figures and/or tables, authored or reviewed drafts of the paper, and approved the final draft.
- Israel Hartuv conceived and designed the experiments, performed the experiments, prepared figures and/or tables, authored or reviewed drafts of the paper, and approved the final draft.
- Uziel Shemesh conceived and designed the experiments, performed the experiments, analyzed the data, prepared figures and/or tables, authored or reviewed drafts of the paper, and approved the final draft.

### Data Availability
Data is available at WorldMeters.info: https://www.worldometers.info/coronavirus/country/china/

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
