# Peer review of "Θ-SEIHRD mathematical model of Covid19-stability analysis using fast-slow decomposition"

_PeerJ, doi:10.7717/peerj.10019_

## Round 0.1 · original submission · Major Revisions

Please address critiques of all reviewers and amend your manuscript accordingly.

Reviewer 1 ·

Basic reporting

I write my comments as an attached file. Please see the attached file.

Experimental design

I write my comments as an attached file. Please see the attached file.

Validity of the findings

I write my comments as an attached file. Please see the attached file.

Additional comments

Review report paper number: Theta SEIHRD mathematical model of Covid19-stability analysis using fast-slow decomposition (#49600).

The paper is very interesting and important from a practical point of view, especially these days and periods.
The paper deal with the application of the singular perturbed vector filed (SPVF) to the theta SEIHRD mathematical model of Covid19. Stability Analysis using a method to decompose the mathematical model into fast-slow subsystems.
Before the publication/ accepted the paper needs a major revision.
The authors (3 authors) should address the following comments to the above paper:
- The abstract is very short and not include all the results presents in the paper. I think that the abstract should include all the main results of the paper.
- What is the meaning of the result: "almost 96 percent"? The authors need to be correct!
- I'm not sure that the title is adequate for the main content of the paper. They should be more specific.
- The introduction is very clear but it seems that it divides into two parts that the connections between these two parts are missing. The first part deal with the review of the COVID-19 and the second part deal with the review of the SPVF. ILDM, CFD, QSSA, etc. which are a method applied to complex ODE and PDE systems. The authors need to connect between these two parts presents in the introduction.
- In addition to the above comment, the introduction includes a subsection of stability analysis. Which also needs to be connected to the above two sections present in the introduction.
- The initial conditions of the mathematical model should be present explicitly.
- The authors present the parameters used in order to solve the mathematical model of COVID 19. But the initial conditions present as a segment and not as one point. So, I ask how the authors solve the model? The need to take one point at each segment of the initial conditions.
- The functions present in equation 17 need an explanation in detail.
- Line 132: " We apply the steps of the algorithm present at 1-8" where is the algorithm.
- If the authors transfer the mathematical model using new coordinates so what are the SMALL parameters of the "new" mathematical model at the new coordinates?
- What is the meaning of the vector present at equation 25?
- The authors claim that the stability of the equilibrium point does not change with the linear transformation. This needs to be proved and present in the paper.
- Line 184: " table ??. At table ??"
- The discussion section needs to be extended.
- The graph present at the paper needs to explain in detail as well as the tables present in the paper.
- The list of references should revise according to the journal format.
- The article should undergo English language editing. There are a few grammatical errors.

Annotated reviews are not available for download in order to protect the identity of reviewers who chose to remain anonymous.

Reviewer 2 ·

Basic reporting

The article is important to predict the number of infected people by covid-19 using mathematical tools. The authors present the mathematical model called Theta SEIHRD model and analyzed the stability of the equilibrium point of the model after the application of the SPVF method.
Why do the authors need to transfer the mathematical model into new coordinates using the eigenvectors of the vector field of the model? They can't investigate the stability of the equilibrium points of the model directly?
The authors should present the graph of the original system as well as the graphs plots of the new model and compared the results.
Nomenclature needs to add the article with its relevant units.
The authors should explain every equation and its expressions and the connections of each dynamical variables of the model.
The abstract need to extend.
Why the authors present subsection of stability analysis at the introductions? This should be a part of the mathematical formulation section.
The algorithm of the SPVF maybe introduces in the article.
The methods ILDM, QSSA, and so on are not relevant for this article since they are asymptotical methods and the SPVF is an algebraic method to transfer the model and present a given mathematical model in new coordinates where at the new coordinates the model can be decomposed into the fast and slow subsystem.
What are the small parameters of the new model?
How the GAP of the eigenvalues indicated the fast and slow system?
The system written in equation (24) is not clear for the reader. Please clarify this system of ODE equations as well as the initial conditions.
The initial conditions of the model are given as a segment. How the initial condition of the model transfer under the eigenvector's transformation and present at the new coordinates?
Equation 29 not clear.
table ??. At table ?? not clear.
Why equation 30 as well as the other equation presents the relative error of the given governing equations.?

Experimental design

The authors didn’t use and prove any results in accordance to the lab. As I understand they are a theoretical researcher. But they can compare their results to experimental results present at the relevant literature.

Validity of the findings

The authors present and compare their results to the numbers that report by the authority of China. The results as present at the Tables the stability analysis of the equilibrium points of the considered model are relevant and they validate their results.

Additional comments

The article is important and can be published at the respectable journal PeerJ but need a major revision. Please take care of my comments very carefully and revise the article in accordance with the PeerJ journal format and criterions.
In addition, the paper needs professional English editing.

Annotated reviews are not available for download in order to protect the identity of reviewers who chose to remain anonymous.

Reviewer 3 ·

Basic reporting

Please see my comments at the file

Experimental design

I attached a file with comments

Validity of the findings

Please see my comments at the file

Additional comments

Please see my comments at the file

Annotated reviews are not available for download in order to protect the identity of reviewers who chose to remain anonymous.

---

## Round 0.2 · Minor Revisions

Please note that in addition to the minor revision in lie with the remaining issue pointed by the reviewers, your manuscript requires editing in English. Therefore you are strongly advised to use help of the professional editor.

Reviewer 1 ·

Basic reporting

Ok

Experimental design

Ok

Validity of the findings

The authors revised the paper

Additional comments

The authors revised the paper but there are still comment that should be addressed before publication:
1: The NOMENCLATURE is not complete
2: The graphs of the governing equation are missing
3: The plot graphs of the new model after that the authors applied the SPVF method are missing
4: The connections between the solution profiles of the model and the "new" model should be present in details at the paper
5: The paper needs English editing.
Good luck

Reviewer 2 ·

Basic reporting

I write down my comments to the editor and the authors

Experimental design

I write down my comments to the editor and the authors

Validity of the findings

I write down my comments to the editor and the authors

Additional comments

The draft of the paper includes all my comments that I wrote previously. There is some issue that should be addressed before the publication of the draft paper:

1: The title should be more correctly specific to China's case. If the authors want to write some general cases they should be compared their results to other countries around the world. Then the paper will be more general.
2: The keywords are generally. They should be more specific to the paper.
3: Units are missing for the Nomenclature.
4: The authors need to explain from where did they get the functions present at Eq. [12]
5: The list of references should be revised according to the journal format.

Reviewer 3 ·

Basic reporting

The authors have corrected the article taking into account my comments.

Experimental design

The authors have corrected the article taking into account my comments.

Validity of the findings

The authors have corrected the article taking into account my comments.

Additional comments

The authors have corrected the article taking into account my comments.

---

## Round 0.3 · accepted · Accept

Thank you for the revised manuscript and response letter. I am pleased to inform you that your manuscript has been accepted for publication in PeerJ.